# Determination of Genetically Identical Strains of Four Honeybee Viruses in Bumblebee Positive Samples

**DOI:** 10.3390/v12111310

**Published:** 2020-11-16

**Authors:** Ivan Toplak, Laura Šimenc, Metka Pislak Ocepek, Danilo Bevk

**Affiliations:** 1Institute of Microbiology and Parasitology, Virology Unit, Veterinary Faculty, University of Ljubljana, Gerbičeva 60, 1000 Ljubljana, Slovenia; laura.simenc@vf.uni-lj.si; 2Institute of Pathology, Wild Animals, Fish and Bees, Veterinary Faculty, University of Ljubljana, Gerbičeva 60, 1000 Ljubljana, Slovenia; metka.pislakocepek@vf.uni-lj.si; 3National Institute of Biology, Večna pot 111, 1000 Ljubljana, Slovenia; danilo.bevk@nib.si

**Keywords:** honeybees, bumblebees, viruses, sequencing, transmission, epidemiology

## Abstract

In recent years, there has been growing evidence that certain types of honeybee viruses could be transmitted between different pollinators. Within a voluntary monitoring programme, 180 honeybee samples (*Apis mellifera carnica*) were collected from affected apiaries between 2007 and 2018. Also from August 2017 to August 2018, a total 148 samples of healthy bumblebees (*Bombus lapidarius, B. pascuorum, B. terrestris, B. lucorum, B. hortorum, B. sylvarum, B. humilis*) were collected at four different locations in Slovenia, and all samples were tested by using RT-PCR methods for six honeybee viruses. Direct sequencing of a total 158 positive samples (acute bee paralysis virus (ABPV *n* = 33), black queen cell virus (BQCV *n* = 75), sacbrood bee virus (SBV *n* = 25) and Lake Sinai virus (LSV *n* = 25)) was performed from obtained RT-PCR products. The genetic comparison of identified positive samples of bumblebees and detected honeybee field strains of ABPV, BQCV, SBV, and LSV demonstrated 98.74% to 100% nucleotide identity between both species. This study not only provides evidence that honeybees and bumblebees are infected with genetically identical or closely related viral strains of four endemically present honeybee viruses but also detected a high diversity of circulating strains in bumblebees, similar as was observed among honeybees. Important new genetic data for endemic strains circulating in honeybees and bumblebees in Slovenia are presented.

## 1. Introduction

Bees are threatened by various pathogens, including viral infections [1]. Among honeybees (*Apis mellifera*), at least 32 different types of viruses have been identified [2]. The most studied are acute bee paralysis virus (ABPV), black queen cell virus (BQCV), chronic bee paralysis virus (CBPV), deformed wing virus (DWV) and sacbrood bee virus (SBV). Regarding the most pathogenic bee viruses, such as ABPV, CBPV, DWV, SBV, and the detection of several novel RNA viruses, studies have been carried out over the previous decade in relation to honeybee losses and the evaluation of their impact on honeybee health [3]. The increased number of investigations of viral infections in honeybee colonies and the use of new technologies, such as the next generation of sequencing (NGS), resulted in the detection of several new RNA and DNA viruses, such as the Lake Sinai virus (LSV) in 2010 [2,3,4,5]. For some of these viruses, it is not yet clear what impact they have on the honeybee colonies, since some viral infections in bees are present as permanent or subclinical infections [6,7,8]. 

In healthy honeybee colonies, the dynamics and the incidence of different viruses over the season can be high. One of the most applied methods of evaluating viral infection in honeybee colony is sampling and testing of randomly selected honeybee workers using RT-PCR [9,10]. In Slovenia, viral infections were detected in previous study with focus on the detection prevalence of ABPV (40%), BQCV (83.3%), CBPV (18.3%), DWV (70%), Kashmir bee virus (KBV = 1.7%) and SBV (8.3%) from 2007 to 2009 in affected honeybee colonies [9]. The simultaneous infection with two to five different viruses was identified as a common pattern, and the number of detected viruses increases with the severity of the clinical condition in the affected honeybee colony [6].

The observed decline of pollinators in the density and diversity on the global scale can also be the result, among other factors, of infection with different viruses. The free-living bumblebees (*Bombus* spp.) are one of the most important pollinators, and they are under the same threat of different factors as honeybees and other species of wild pollinators [6]. In a study conducted in the United Kingdom at 26 locations, it has been shown that four bee viruses (BQCV, DWV, ABPV and slow bee paralysis virus (SBPV)) have been detected in honeybees and bumblebees. By using the quantitative RT-PCR methods from positively detected *A. mellifera* and *Bombus* spp. for BQCV, virus loads were not significantly different (10^4^–10^6^ virus particles), while for forager *A. mellifera* samples the viral load was much greater (10^10^–10^11^ virus particles) than in *Bombus* spp. In the same study, it was also determined that the frequency and occurrence of a single virus in honeybees is a good indicator of the presence of this virus in bumblebees [11]. A study performed in Vermont, USA, showed that the prevalence of DWV and BQCV were higher in bumblebees collected, near apiaries and when neighbouring honeybees had high infection levels [12]. 

Honeybee viruses can be transmitted horizontally and vertically [13,14]. Trans-species transmission of different pathogens has also been well known and studied for a long time in animal and human pathology [15]. Therefore, genetic studies of honeybee viruses, including the identification of trans-species transmissions, provide important data for understanding the epidemiology of individual viruses. Some viruses may be present in different species living in the environment closely with honeybees and meet through the direct or indirect contacts. The exact data about the susceptibility and ways of transmission of honeybee viruses between different types of pollinators remains very limited. The current understanding suggests the high complexity of viral infections, and the results may be difficult to interpret due to the interactions between various pollinators. The hypothesis that viruses are spilling over from managed honeybees to wild bumblebees and that contaminated flowers may be an important route for transmission is highly probable [11,12]. A recent phylogenetic analysis of honeybee and varroa mite (*Varroa destructor*) samples in Slovenia confirmed that genetically identical strains of DWV were detected in both species and that varroa is an efficient vector between honeybee colonies and apiaries [16].

With the RT-PCR method testing of four samples of dead bumblebees (*Bombus humilis, Bombus pascuorum, Bombus hypnorum* and *Bombus lapidarious*) collected in 2017, the presence of nucleic acids of two honeybee viruses (ABPV and BQCV) was demonstrated for the first time in Slovenia [17]. From October 2016 to December 2017, 56 honeybee samples from 32 different locations and 41 bumblebee samples from five different locations were collected. The result of testing by reaction (RT-PCR) method showed that and 75.92% of honeybee samples and 17.07% of bumblebee samples were LSV positive. The identified strains of LSV from bumblebees share from 98.6% to 99.4% nucleotide identity to the closest honeybee LSV strains collected from the same geographical area in Slovenia [5]. LSVs have also been also detected, with similar incidences and titres in historical European honeybee samples. LSV-1 and LSV-2 have strong similarities in capsid and genome size, seasonal incidence, predominantly adult based infection and absence of overt symptoms with Bee virus Y and Bee virus X respectively and may therefore be related [18].

In this comprehensive study, the detection and phylogenetic comparison of ABPV, BQCV, SBV and LSV positive samples was performed for the first time in Slovenia with honeybee (*Apis mellifera carnica*) and bumblebee (*Bombus* spp.) samples to determine whether identical strains of honeybee viruses could also be detected in bumblebees.

## 2. Materials and Methods 

Within this research, samples were collected in two frameworks. In the first, from 2007 to 2018, 180 samples of honeybee workers (10 honeybees in each sample) were collected within voluntary monitoring programme by the veterinarian specialists for bee health from clinically affected or dead colonies. Samples were randomly selected from the whole territory of Slovenia and stored frozen at less than minus 50 °C in the laboratory until use. In the second, samples of individual bumblebees were collected in August 2017 and August 2018 within a project (CRP V4-1622) financed by the Slovenian Research Agency of the Republic of Slovenia. A total 148 clinically healthy individual bumblebees of species *Bombus lapidarius* (BL), *B. pascuorum* (BP), *B. terrestris/lucorum* (BT) *B. hortorum* (BHO)*, B. sylvarum* (BS), *B. humilis* (BHU) were sampled on flowers at four different locations (named Location 1, 2, 3 and 4), which were from 50 to 100 km apart (Figure 1). At the same day, when individual bumblebees were sampled, ten clinically healthy honeybee workers (*Apis mellifera carnica*) were collected at each location and a total eight honeybee samples on four locations (Table 1). All samples were frozen and stored below minus 50 °C until the start of the investigation. The honeybee samples had a protocol number and the year of sample collection noted (e.g., 246/2016), whiles for bumblebee samples bumblebee species, sampling number and year of sample collection were noted (e.g., Bombus-BL3/2017). 

At first, a pool of ten honeybees was placed into Ultra-Turrax DT-20 tubes (IKA, Königswinter, Germany). This pool represented one sample of collected honeybees from each sampling time or location. Then, 5 mL of RPMI 1640 medium (Gibco, Paisley, UK) were added to each sample, and they were incubated at room temperature for 30 min. Similarly, samples of individual bumblebee were prepared by adding 3 mL of RPMI 1640 medium (Gibco, UK) into a tube, and each collected sample of bumblebee was tested individually. Samples were homogenised and centrifuged for 5 min at 2500× *g*. The total RNA was isolated from individual samples with the QIAamp viral RNA mini-kit (Qiagen, Hilden, Germany) according to the manufacturer’s instructions.

Nucleic acids of five honeybee viruses (ABPV, BQCV, CBPV, DWV and SBV) were detected using a specific method of reverse transcription and polymerase chain reaction (RT-PCR), as previously described [6]. The LSV was detected with a specific RT-PCR method [5] using previously described primers LSV 1765-F and LSV 2368-R [19]. Results were evaluated based on the size of RT-PCR products in the agarose gel as positive in the case of the expected product size: for ABPV 452 nucleotides (nt), for BQCV 770 nt, for CBPV 570 nt, for DWV 504 nt, for SBV 814 nt and for LSV 603 nt.

In the case of a positive result, the selected RT-PCR products of a single virus were directly sequenced with the Sanger sequencing protocol, using the same primers as used for specific RT-PCR as described previously [5,9,16]. Individual sequences were analysed using the DNASTAR 5.05 (Lasergen, WI, USA) programme and 158 positive samples of four viruses (ABPV: *n* = 33, BQCV: *n* = 75, LSV: *n* = 25 and SBV: *n* = 25) detected in honeybees and bumblebees together with closely related sequences from GenBank and interpreted according to the results of the nucleotide sequence matching between honeybee and bumblebee samples. Multiple alignments were created using program MEGA 6.06. Genetic distances were calculated from the alignment based on the Tamura 3-parameter model, and phylogenetic trees were generated using the Maximum Likelihood (ML) statistical method implemented with the Tamura 3-parameter model with Gamma distribution [20]. The test of phylogeny was performed through 1000 bootstrap replicates. Only bootstrap values higher than 70% were presented on phylogenetic trees. 

The comparative analyses of collected samples at four locations, comparison with previously detected viruses in honeybees and with the most closely related sequences available in GenBank were performed.

## 3. Results

### 3.1. The Prevalence of ABPV, BQCV, CBPV, DWV, SBV, and LSV in Bumblebee Samples

A total of 148 bumblebee samples were tested by specific RT-PCR methods and evaluated for detection of six honeybee viruses. From tested samples, 25 (37.8%) of 66 collected samples in 2017 were totally negative, and 9 (10.9%) of 82 in 2018 were negative for six tested honeybee viruses. The RT-PCR results in six species of bumblebees (*B. lapidarius*, *B. pascuorum*, *B. terrestris/lucorum*, *B. hortorum, B. sylvarum, B. humilis*) showed various prevalence, from 0% to 100% for six honeybee viruses were detected (Table 1). Of the 148 tested samples of bumblebee, 8.8% (*n* = 13), 58.1% (*n* = 86), 0% (*n* = 0), 2.7% (*n* = 4), 24.3% (*n* = 36) and 14.8% (*n* = 22) were detected positive for ABPV, BQCV, CBPV, DWV, SBV and LSV, respectively. More than 50 per cent of collected honeybee samples were positive for ABPV, BQCV, SBV, and LSV at the same times and locations as the bumblebee samples were collected. Lower percentages of positive samples were detected in bumblebees than in honeybees for all six viruses (Table 1, Appendix A). High differences in the prevalence were observed for individual viruses, between different viruses in bumblebees and samples collected in August 2017 and 2018 (Table 1).

Honeybee viruses were detected with high variability in different bumblebee species, except for CBPV, which was not detected in any of the collected samples of bumblebees in 2017 and 2018 (Table 1). The highest percentage of positive samples in bumblebees was detected for BQCV (80.49% in 2018), while the second most prevalent virus was SBV with 27.27% in 2017, followed by LSV with 19.51% in 2018, ABPV with 13.64 in 2017 and DWV with 4.55% in 2017. In comparing the prevalence of positive *Bombus* spp. samples in 2017 with results obtained in 2018 for individual viruses, higher prevalences were detected for ABPV, SBV, and DWV in 2017, while higher prevalences were detected for BQCV and LSV in 2018 than in 2017 (Appendix A).

### 3.2. Detection of Genetically Identical Strains of ABPV, BQCV, SBV, and LSV in Honeybee and Bumblebee Positive Samples

From positive samples of honeybee workers, collected from affected colonies between 2007 and 2018, a total 117 new sequences (named in Appendix A as “monitoring” samples) were determined (ABPV = 28, BQCV = 60, SBV = 9 and LSV = 20), representing clinically affected colonies from the whole country and the national collection of circulating strains of honeybee viruses during last decade in Slovenia. From RT-PCR, positive samples of honeybee workers (eight positive samples) and bumblebees (individual), collected from four locations (Locations 1–4), a total 41 sequences were determined (ABPV = 5, BQCV = 15, SBV = 16 and LSV = 5) (Appendix A). The comparison of ABPV, BQCV, SBV, and LSV positive samples revealed from 98.74% to 100% nucleotide identity between collected honeybee and positive bumblebee samples (Table 2). 

During sampling of bumblebee samples in August 2017 on Location 3, 80% of tested samples were detected SBV positive. At the same location, the highest percentage (58.8%) for SBV-positive samples was also detected in 2018, when comparing the prevalence in all four locations (Appendix A). The genetic comparison of honeybee and bumblebee positive samples collected at the same time and on the same location have shown high nucleotide identity among BQCV, SBV, and LSV. Also, high genetic diversity among sequenced positive samples was detected for BQCV, collected on the same location (not presented). The comparison was possible only for those positive and sequenced samples, which were detected in both species at the same location (Appendix A).

The genetic comparisons of ABPV positive samples showed that 100% nucleotide identity was observed for honeybee strain ABPV 281/2016 (MH900044) and Bombus-BL3/2017 (MH900049); 99.75% for ABPV honeybee strain 246/2016 (MH900021) and Bombus-BP4/2017 (MH900051) and 99.26% nucleotide identity between honeybee strain ABPV M99/2010 (HQ877404) collected in Slovenia in 2010 and Bombus-BT-29/2017 (MH900052) (Table 2, Figure 2).

Genetically 100% nucleotide identical BQCV strains were found in bumblebees and detected honeybee samples BQCV 279/2017 (MH899977) and Bombus-BT23/2017 (MH900014); BQCV 1960-1/2009 (MH899996) and Bombus-BT27/2017 (MH900015); BQCV 287-2/2017 (MH899978) and Bombus-BL-6/2017 (MH900010) (Table 2). The identified BQCV positive samples from bumblebees were located on several branches on the phylogenetic tree, along with honeybee BQCV positive samples (Table 2, Figure 3). With presenting only the most closely related strains from the same location, BQCV strain LJU/2017 (MH900004) had 99.69% nucleotide identity with BQCV strain Bombus-BP23/2017 (MH900013), collected from the same day (28 August 2017) on Location 4, while BQCV strain Bombus-BP23/2017 (MH900013) had 100% nucleotide identity with another honeybee strain GRM/2017 (MH900004), collected on Location 1 (70 km apart from Location 4) on 9 August 2017. The detected honeybee strain BQCV NAK/2017 (MH900006) collected on Location 3 had 99.69% nucleotide identity with BQCV strain Bombus BP30/2017 (MH900017), which was also collected on Location 3 (Appendix A, Figure 3).

Several strains of SBV detected in positive bumblebee samples were closely related to each other and to positive honeybee samples, located on two separate branches (Figure 4); 100% nucleotide identities were detected between SBV NAK/2017 (MH900064) and Bombus-BP11/2017 (MH90065) and a group of strains represented by SBV AZ2/2016 (MH900059) and Bombus-BT33/2017 (MH90078) (Table 2, Figure 4). Honeybee SBV strain LJU/2017 (MH900060) has 99.87% nucleotide identity with SBV strain Bombus-BT33/2017 (MH900078), both collected on the same day (28 August 2017) from Location 4. The detected honeybee strain SBV NAK/2017 (MH900064) collected on Location 3 had 100% nucleotide identity with twelve SBV strains detected in bumblebees (MH900065-MH900077), collected at the same day on Location 3 (Appendix A, Figure 4).

The most closely related honeybee positive sample LSV3M92/2010 (MG918125), collected in 2010 have 99.28% nucleotide identity to LSV3/BombusBT35/2017 (MH350871), while between honeybee strain LSV3/CBNA/2017 (MH350882) and bumblebee strain LSV3/BombusBP9/2017 98.74% identity was detected. High (99.64%) nucleotide identity was observed between LSV2/286/2017 (MH350890) and LSV3/BombusBP21/2017 (MH350889), confirming the identification of genetically closely related strains of LSV among honeybee and bumblebee (Table 2, Figure 5). Honeybee LSV3 CBLJ/2017 (MH350878) had 98.20% nucleotide identity with LSV3 strain Bombus BT35/2017 (MH900071), both collected on the same day (28 August 2017) from Location 4 (Appendix A, Figure 5).

### 3.3. Phylogenetic Comparison of Detected Strains of ABPV in Honeybee and Bumblebee Samples

The phylogenetic comparison of detected 25 positive honeybee samples and eight positive bumblebee samples of ABPV (GenBank accession numbers MH900021–MH900053) from this study showed that several ABPV strains, detected from both species, were genetically closely related to each other and also closely related to previously detected ABPV strains in Slovenia. Comparison of the 408-nucleotide long sequence of the ORF1 region of the viral genome showed that identical ABPV strains were detected in positive samples of bumblebees, with 99.3% to 100% nucleotide identity to the closely related ABPV strains in honeybees (Figure 3). Among positive samples, detected in the *B. lapidarius, B. pascuorum* and *B. terrestris/lucorum*, the presence of at least three different lineages of ABPV strains were identified with divergence from each other in the ORF1 genome region by up to 1.5% at the nucleotide level.

The BLAST comparison of a representative strain of ABPV 246/2016 (MH900021) with the most closely related strains in GenBank showed 99.2% nucleotide identity with strain Hungary 1 (AF486072), 98.77% identity with strain Am20 from the United Kingdom and 97.06 with strain Poland 1 (AF486073). The detected 99.51% of nucleotide identity between ABPV strains from this study and strain M37/2010 (HQ877406) revealed that the same strain of ABPV was detected in 2016 and 2017 at distant locations in Slovenia and is the result of successful transmission of honeybee ABPV strains during the previous decade (Figure 2).

### 3.4. Phylogenetic Comparison of Detected Strains of BQCV in Honeybee and Bumblebee Samples

A phylogenetic comparison of 63 BQCV positive honeybee samples and 12 positive samples from bumblebees showed high genetic diversity of detected strains of BQCV with a maximum 4.1% difference at the nucleotide level. The identified bumblebee strains of BQCV were located on several branches together with honeybee strains of BQCV collected within monitoring programme between 2007 and 2018 (Figure 3). A comparison of the 653 nucleotide-long sequences of the capsid protein also showed that genetically identical or closely related strains of BQCV were detected in honeybees and bumblebee, with 98.5% to 100% nucleotide identity. 

The genetic comparison of 75 BQCV positive samples collected between 2007 and 2018 showed that at least 15 different BQCV strains were identified in Slovenia. The most closely related strain to the detected honeybee BQCV 286/2007 (MH899982) was BQCV/21/Serbia/2015 with 100% nucleotide identity. The closely related were BQCV/144l (MN565034) strain with 99.23% nucleotide identity recently detected in *Vespa velutina nigrithorax* in France and strain CP (HG764796) from *Apis mellifera* detected in 2012 in Belgium. Another group of detected Slovenian BQCV strains had 99.39% nucleotide identity with strains A (KT152152) and 99.08% identity with Australian strain SA (KY465685). The Slovenian BQCV strain 1930-1/2009 (MH899995) collected from honeybee samples in 2009 have been one of the most divergent of the detected strains, with 97.55% nucleotide identity to the closest strain Bombus-290/2017 (MH900007).

### 3.5. Phylogenetic Comparison of Detected Strains of SBV in Honeybee and Bumblebee Samples

The phylogenetic comparison of SBV-positive samples with determined 783-nucleotide-long sequences of 11 positive honeybee samples and 14 positive samples of bumblebees showed the circulation of at least three different strains of SBV in Slovenia. The nucleotide comparison of genome region, encoding the polyprotein, showed a 100% nucleotide identity between the positive samples of SBV strains detected in honeybee and bumblebee samples (Figure 4, Table 2). The positive SBV samples of honeybees and bumblebees sampled at the same places (Locations 3 and 4) had 100% nucleotide identity, which was evidence that the same strain of SBV was detected in honeybees and bumblebees at the time of sampling.

To the identified 25 positive samples of SBV from Slovenia, the most closely related strains in GenBank were strain MD1 (MG545286) with 97.96% nucleotide identity from the US and 97.83% identity with the Lithuanian strain 45-13 (KP223786). The most divergent SBV 153/2009 (MT900056) strain in this study was collected in 2009 from an affected honeybee brood sample and had only 89.02% nucleotide identity with closest strain in GenBank (SBV/PNG-2018, MT482478), confirming the identification of new strain of SBV, which was not detected in any of other 24 samples in this study.

### 3.6. Phylogenetic Comparison of Detected Strains of LSV in Honeybee and Bumblebee Samples

A comparison of the 557-long nucleotide sequences of the genome region coding for the LSV RNA polymerase virus, with 22 positive honeybee samples and three positive bumblebee samples, showed a high diversity of detected LSVs, with a circulation of at least four different LSV strains in Slovenia, as well as the identification of closely related strains of LSV on each detected lineage among honeybee and bumblebees (Figure 5). 

## 4. Discussion

This research confirmed that genetically the same strains of four honeybee viruses (ABPV, BQCV, SBV and LSV) were also identified among bumblebees, collected at the same time and at the same location (geographic area). Our findings are also valuable for understanding previous observations that several honeybee viruses may be detected by RT-PCR methods in other pollinators, mixing with wild pollinators during foraging on shared flower resources [7,8,9,10,11,12,21,22]. The detected prevalence among tested 148 bumblebee samples was lower than observed prevalence during the same period in honeybees (Table 1). We are aware that pooled samples of ten bees are more likely to be positive than that of a single bee since only a few bees out of ten need to be positive for the whole sample to be positive. The positive rates for tested honeybee samples may be overestimated compared to those results for single bumblebee testing, but in this study, the positive samples were mostly used in a descriptive way and for phylogenetic studies and thus had little effect on the conclusions. In contrast, the detected average 24.3% prevalence among tested bumblebees for SBV was higher than expected but is related mainly to bumblebee samples from Locations 3 and 4, where 40% to 80% of tested samples were positive (Appendix A). The high detection rate of SBV infection (and also other viruses) in bumblebees can be also critical for managed honeybees as a possible source of SBV infection because the same strain was also detected in healthy honeybee samples collected at the same location as the control group. This observation is strongly related to geographic area and virus in the study, which is supported by a previous study on BQCV and DWV prevalence in the USA among wild bees, where BQCV was detected extremely rarely in wild bees, while DWV was very common across all groups [23]. The current study also provides clear evidence that several honeybee viruses can be detected in any of the six tested species of bumblebees, supporting observations of some previous studies for finding possible causes for the decline of pollinators [6,18]. The detected variations in prevalence were observed in the same locations when comparing the results of tested individual bumblebee samples in 2017 and 2018 for different viruses. Only clinically healthy bumblebees were collected and tested in this study; to date, it is not clear if these field viruses could cause a clinical form of the disease in bumblebees, as is described in honeybees [9,24]. 

CBPV was not detected in any of the tested bumblebee samples from this study, which may be due to the sampling strategy, as only healthy bumblebees were collected from flowers. Only four samples of tested bumblebees were DWV positive (2.70%), confirming the low prevalence of DWV among healthy bumblebees. That is not surprising, because this virus is strongly associated with geographic area, time of sampling and honeybees varroa mite infestation, and it is detected in much higher percentages in clinically affected bees and colonies. For the CBPV and DWV, it is characteristic that nearby subclinical infections of bees they can also cause clinical changes in individuals or in a larger number of honeybees (deformed wings, paralysis) in diseased honeybee colonies. Because of this low prevalence, the phylogenetic comparison for detected strains of CBPV and DWV in honeybee and bumblebee was not made in this study. A sampling of affected bumblebees would be very difficult, since wildlife species, including bumblebee species, die in their environment [17]. 

This comprehensive study was based on genetic comparison of 158 positive samples of ABPV, BQCV, SBV and LSV, which were sampled and detected within affected honeybee colonies between 2007 and 2018 and from healthy bumblebees collected on flowers at four locations in 2017 and 2018. We found that both honeybees and bumblebees were infected with genetically identical or very closely related honeybee strains (98.74% to 100% nucleotide identity in sequenced regions) of ABPV, BQCV, SBV and LSV. High sequence similarity between detected field strains is direct evidence of the spillover of these viruses, similar as previously has been suggested for DWV between different types of pollinators [6,11,17,23,24,25]. Our study also confirmed that genetically identical strains were identified in bumblebees when comparing the results to strains detected in positive honeybee samples collected with a long-existing voluntary monitoring programme in Slovenia. This was direct confirmation that several endemic strains, detected in honeybees in our territory are also infecting bumblebees. The comparison with the most closely related strains in GenBank revealed the detection of new viral strains and previously detected endemically present strains in Slovenia and other countries [10,26]. The second important observation in this study was the identification of high genetic diversity among circulating strains of four sequenced types of honeybee viruses on the relatively small geographic area of Slovenia. The observed diversity among bumblebees was like those detected in honeybee positive samples, confirming that genetically different strains of bee viruses are infecting both species.

A series of experimental infections have shown that three viral types isolated from honeybees (DWV genotype A, DWV genotype B and BQCV) readily replicate within hosts of the bumblebee *B. terrestris*. Impacts of these honeybee-derived viruses—either injected or fed—on the mortality of *B. terrestris* workers were negligible and probably dependent on host condition [27]. The identification and sequencing of an individual strain of honeybee viruses in bumblebee cannot be directly linked to the possible pathology but help us to understand the spillover events between different species. With experimental infection studies, acute virulence of individual honeybee viruses in bumblebees has been proved and is correlated with the dose of individual viruses. By using the buff-tailed bumblebee, a generalist forager in the Palearctic region, it has been previously demonstrated that oral feeding of 0.5 × 10^7^ and 1 × 10^7^ viral particles per bee of either Israeli acute paralysis (IAPV) or Kashmir bee virus, respectively, resulted in an active infection and fitness loss, while a dose of 0.5 × 10^6^ IAPV particles per bee was not infectious [28]. Experiments elsewhere showed that oral infection of *B. terrestris* workers with 10^9^ genome copies of a different honeybee virus, DWV, reduced the mean survival of *B. terrestris* workers by six days [6]. Nevertheless, when detecting genetically different field strains of viruses and the observed virulence data for only some of these viruses is known, the direct conclusions should not be generally accepted.

The honeybees could be infected with some virus because of close contact with infected bees from other colonies, contaminated flowers, infected varroa mite, bumblebees, or other wild bees [11]. In a previous study in which samples were collected during a 14-month period of one season within the same honeybee colonies and apiaries, we determined that ABPV, BQCV, CBPV, and DWV viruses were the most frequently detected in infected foraging bees and in a lower percentage in house bees, while the lowest percentage was found in the pupae samples [10]. This finding confirms that in most cases foraging bees could be responsible for the transmission of viruses between colonies and therefore there is a possibility for spreading of honeybee viruses during pasture. As was previously confirmed for DWV by sequence comparison of detected strains, this honeybee virus is probably horizontally transmitted within the honeybee colony and between colonies through direct or indirect contact among the foraging and hive honeybees or by infected varroa mites [16,26]. The less frequent transmission avenue is vertical, where the source of infection is the queen [13,18]. 

Although the quantification of positive samples of ABPV, BQCV, SBV, and LSV was not possible because of the limitation of RT-PCR methods used in our study, and thus viral load was not determined, some previous studies have revealed that viral loads in *Bombus* spp. or wild bees are lower than in *A. mellifera* foragers [11,23]. The evidence, that same viral strains were found in six species of bumblebees with some variations in the prevalence during 2017, and 2018 revealed that identified strains are not specific for individual species. Due to the relatively small number of examined and compared virus-positive samples, we cannot conclude that all tested species of bumblebees are equally susceptible to infection with an individual honeybee virus. Although most of these viruses were historically first detected in honeybees, that does not mean that honeybees are the only or even primary host. Perhaps it would be more accurate in the future to call these ‘‘bee viruses’’ which would fit better with both our current understanding of virus infections in different bee species, because calling them ‘‘honeybee viruses’’ sounds too species-specific. 

In comparison with a previous study in the United Kingdom, in which only four types of viruses were found by RT-PCR [11], but not further genetically confirmed with sequencing, our phylogenetic study gives a much more accurate insight into the diversity of ABPV, BQCV, SBV, and LSV and the prevalence of honeybee viruses in bumblebees. The identification of genetically closely related honeybee viruses in bumblebees in collected samples from the same geographic area was confirmed by sequencing detected strains. In any case, for the better control of viral infections in honeybees, such studies should be extended to other types of pollinators. Through this approach, we will obtain a more complete picture of the actual prevalence of different strains of viruses in the environment. Our new understanding of the role of viruses in mutualistic symbiotic relationships with their hosts is expanding as our knowledge of the virome, through sequencing technologies and bioinformatic strategies are rapidly increasing. With an increasing number of detected and sequenced honeybee viruses, some aspects of how these viruses might be fused with their hosts in symbiogenetic relationships should also be considered. Viewing detected virus infections in the context of ecology provides a framework for a deeper understanding of the intertwined relationships of all life, including viruses. It is inevitable that more examples of mutualistic viruses will be revealed as we continue this exciting phase of different bee virus discovery.

## 5. Conclusions

This study confirms that several positive-stranded RNA viruses, found in managed honeybees (*Apis mellifera carnica*) may represent a complex of emerging infectious viral diseases, as a part of the global problem of decreasing pollinators. From this study, we cannot conclude if viral infection spills over from honeybees to bumblebees, or vice versa, but it is most likely that these viruses could be spreading in both directions.

## Figures and Tables

**Figure 1 viruses-12-01310-f001:**
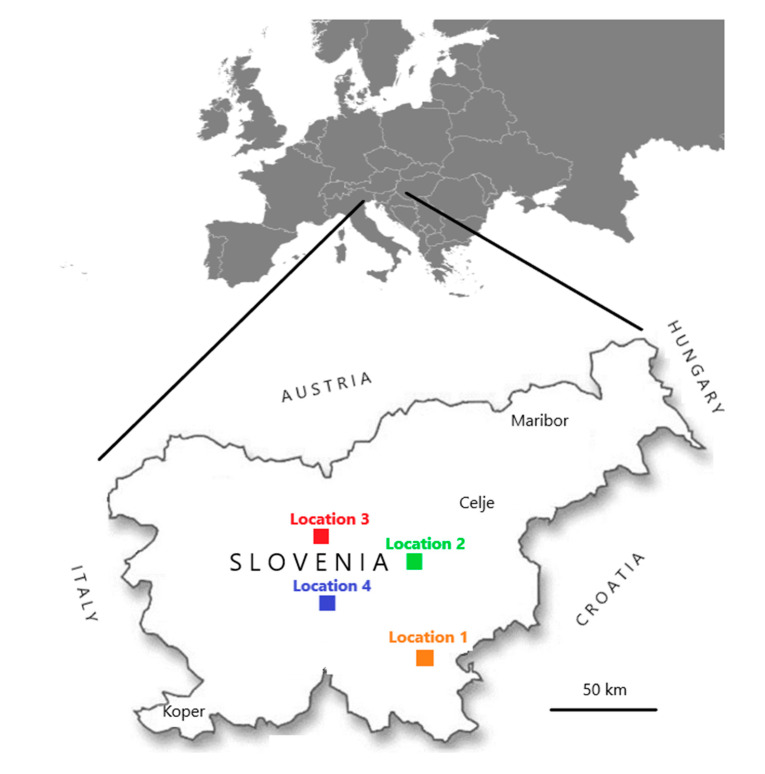
Four places (named Locations 1, 2, 3, and 4) in Slovenia where a total of 148 clinically healthy individual bumblebees and eight honeybee samples were collected during August 2017 and August 2018.

**Figure 2 viruses-12-01310-f002:**
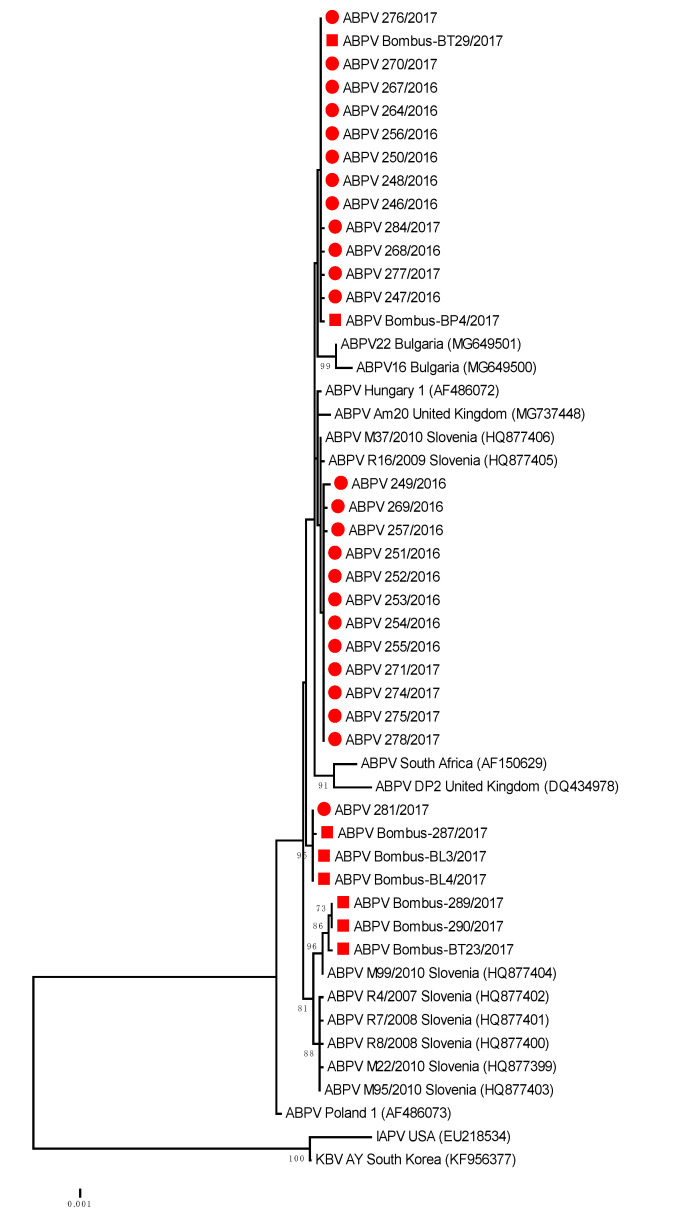
Phylogenetic comparison of 408 nucleotide-long sequences of ORF 1 (RNA-dependent RNA polymerase gene) of ABPV (nt genome position between 5.281 and 5.688, numbering according to strain Hungary 1, AF486072) for 25 positive samples obtained from honeybees (●) and eight positive bumblebee samples (■) in this study. The maximum likelihood tree is presented for 33 Slovenian ABPV samples with 17 strains from GenBank (with accession numbers).

**Figure 3 viruses-12-01310-f003:**
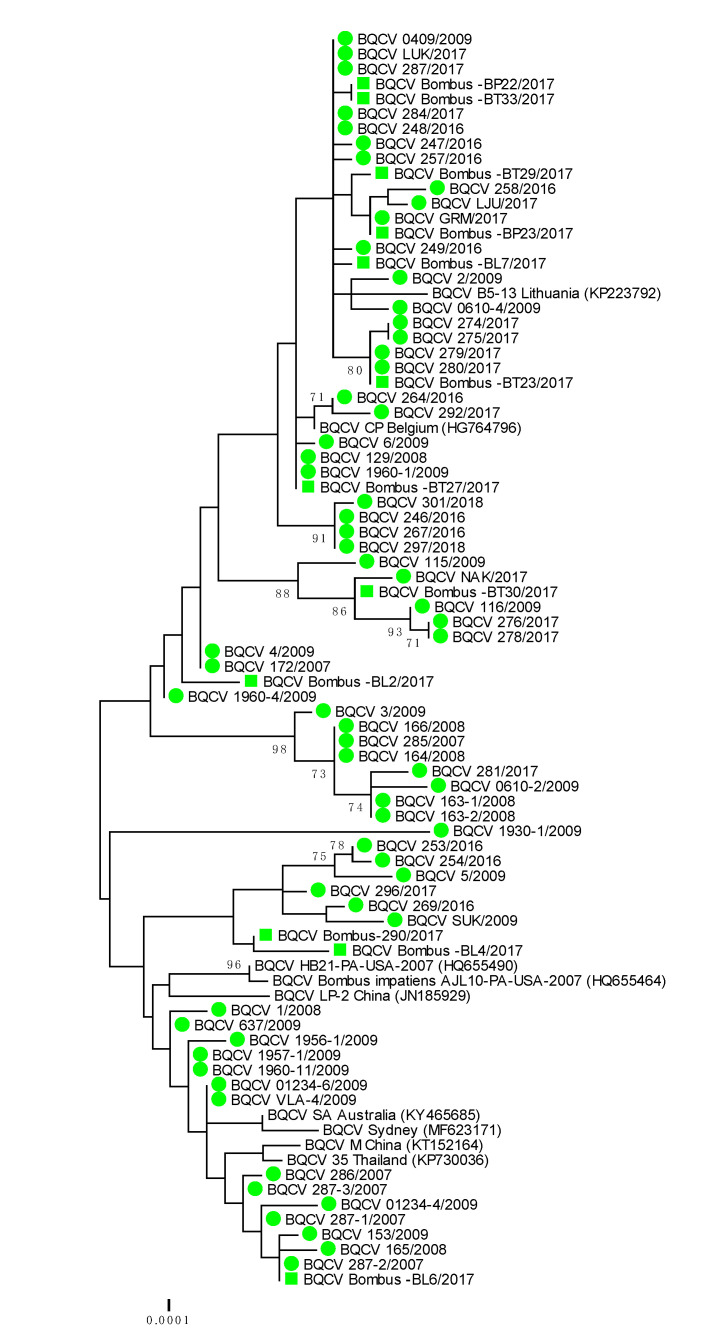
Phylogenetic comparison of 63 BQCV positive samples of honeybees (●) and 12 positive bumblebee samples (■) in Slovenia (nt genome position between 7788 and 8440, numbering according to strain BQCV/144l, MN565034). The comparison was performed on the determined sequence of the 653 nucleotides of the viral genome region encoding the capsid protein together with the nine genetically closest strains detected in GenBank (presented with accession numbers).

**Figure 4 viruses-12-01310-f004:**
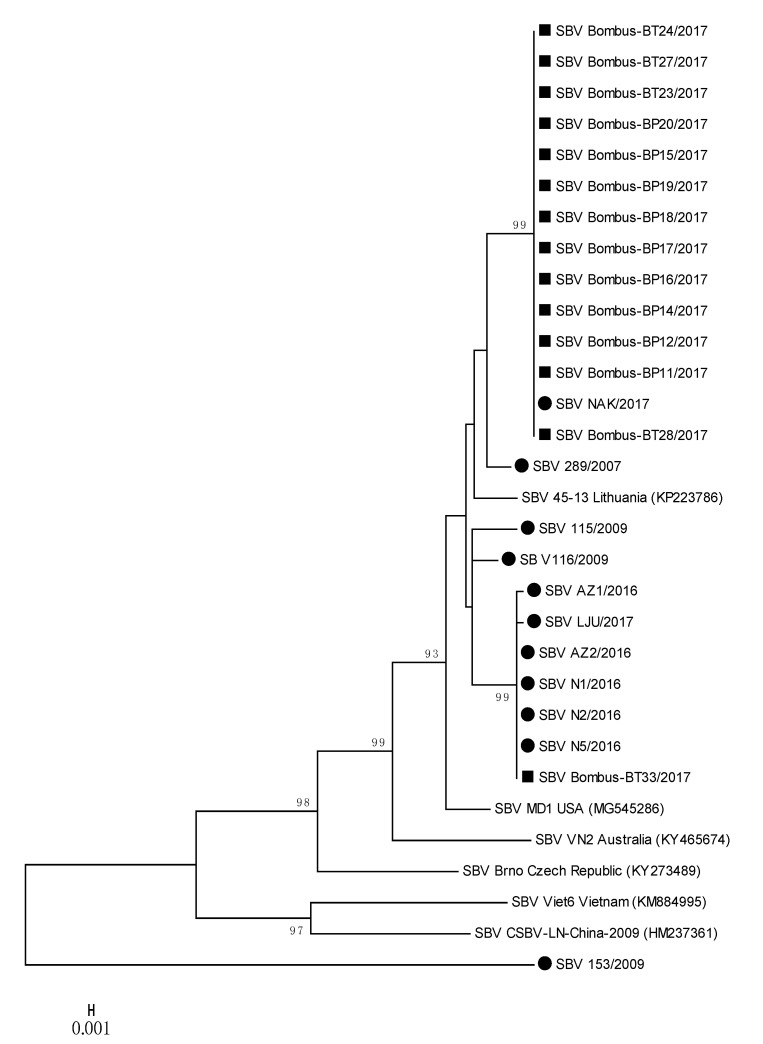
Phylogenetic comparison of 11 SBV positive samples detected in honeybees (●) and 14 SBV positive samples from bumblebees (■) in Slovenia. The comparison was based on the 783-nucleotide long sequences of the viral genome region coding for polyprotein (nt genome position between 4992 and 5774, numbering according to strain MD1, MG545286), together with the six genetically closest genes from the GenBank (with accession numbers).

**Figure 5 viruses-12-01310-f005:**
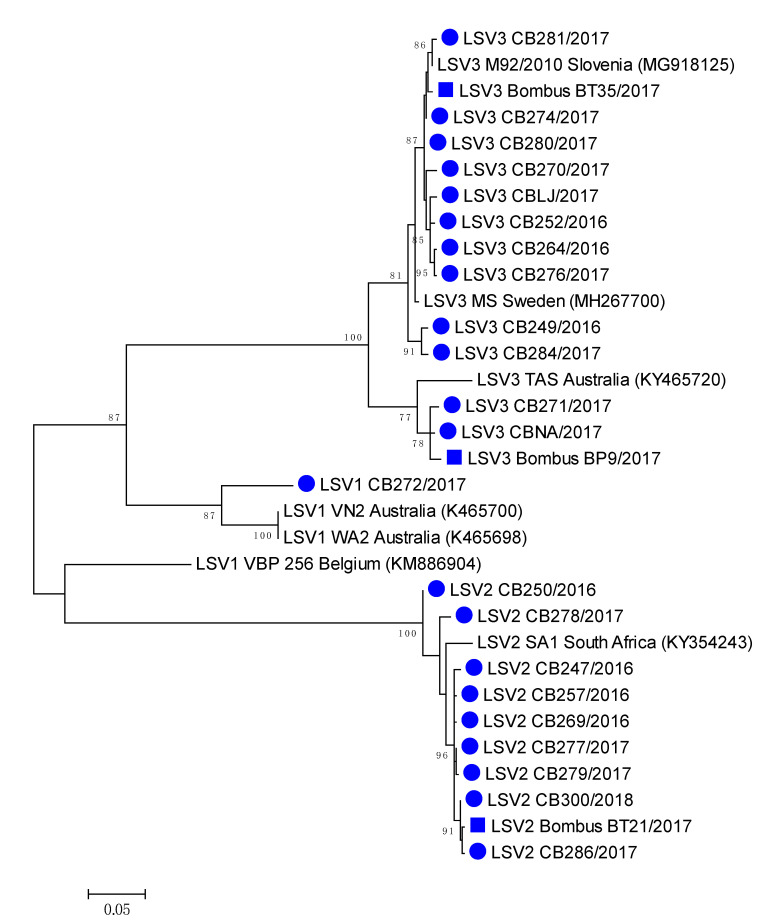
Phylogenetic comparison of the 22 LSV-positive in honeybees (●) and three bumblebee (■) samples in Slovenia. The comparison was performed based on the 557 nucleotide sequences of the partial viral genome region encoding the RNA polymerase (nt genome position between 1778 and 2333, numbering according to strain LSV2/SI, MT482482), together with the seven genetically closest strains detected in GenBank (presented with accession numbers).

**Table 1 viruses-12-01310-t001:** The percentage of detected *Bombus lapidarius*, *B. pascuorum*, *B. terrestris/lucorum B. hortorum, B. humilis, B. sylvarum and Apis mellifera* positive samples for ABPV, BQCV, CBPV, DWV, SBV and LSV by specific RT-PCR methods, collected in August 2017 and 2018.

Species	No of Samples	Year of Sampling	ABPV	BQCV	CBPV	DWV	SBV	LSV
*B. lapidarius*	*n* = 10	2017	30%	60%	0%	10%	0%	0%
*B. pascuorum*	*n* = 26	2017	7.69%	23.1%	0%	0%	30.8%	11.5%
*B. terrestris/lucorum*	*n* = 30	2017	13.3%	26.3%	0%	6.6%	33.3%	10%
*Apis mellifera*	*n* = 4	2017	75%	100%	0%	25%	50%	75%
*B. lapidarius*	*n* = 11	2018	0%	90%	0%	0%	0%	9.1%
*B. pascuorum*	*n* = 29	2018	3.4%	65.5%	0%	0%	48%	24.1%
*B. terrestris/lucorum*	*n* = 25	2018	4%	84%	0%	0%	4%	28%
*B. hortorum*	*n* = 3	2018	66%	100%	0%	0%	33%	33%
*B. humilis*	*n* = 2	2018	0%	100%	0%	50%	0%	0%
*B. sylvarum*	*n* = 12	2018	0%	91.7%	0%	0%	16.7%	8.3%
*Apis mellifera*	*n* = 4	2018	50%	100%	25%	25%	50%	100%
Average *Bombus* spp.	*n* = 148	2017–2018	8.8%	58.1%	0%	2.7%	24.3%	14.8%
Average *Apis mellifera*	*n* = 8	2017–2018	62.5%	100%	12.5%	25%	50%	87.5%

ABPV, acute bee paralysis virus; BQCV, black queen cell virus; CBPV, chronic bee paralysis virus; DWV, deformed wing virus; SBV, sacbrood bee virus; LSV, Lake Sinai virus.

**Table 2 viruses-12-01310-t002:** The representatives of most closely related strains with high nucleotide identity observed for ABPV, BQCV, SBV and LSV among bumblebees (*Bombus* spp.) and honeybees (*Apis mellifera*) positive samples.

	Honeybee Samples		Bumblebee Samples			
Honeybee Virus	Name of Honeybee Sample *(Date of Sampling)*	GenBank Accession Number	Name of Bumblebee Sample *(Date of Sampling)*	GenBank Accession Number	Number of Nucleotides Comparison	% of Nucleotide identity between Honeybee and Bumblebee Positive Samples
ABPV	246/2016 *(7 October 2016)*	MH900021	Bombus-BP4/2017 *(9 August 2017)*	MH900051	408	99.75%
ABPV	281/2016 *(10 May 2017)*	MH900044	Bombus-BL3/2017 *(9 August 2017)*	MH900049	408	100.00%
ABPV	M99/2010 *(17 May 2010)*	HQ877404	Bombus-BT29/2017 *(23 August 2017)*	MH900047	408	99.26%
BQCV	279/2017 *(21 April 2017)*	MH899977	Bombus-BT23/2017 *(22 August 2017)*	MH900014	653	100.00%
BQCV	1960-1/2009 *(26 August 2009)*	MH899998	Bombus-BT27/2017 *(22 August 2017)*	MH900015	653	100.00%
BQCV	287-2/2007 *(14 June 2007)*	MH899984	Bombus-BL6/2017 *(9 August 2017*)	MH900010	653	100.00%
SBV	NAK/2017 *(10 August 2017)*	MH900064	Bombus-BP11/2017 *(10 August 2017)*	MH900065	783	100.00%
SBV	AZ2/2016 *(14 June 2016)*	MH900059	Bombus-BT33/2017 *(28 August 2017)*	MH900078	783	100.00%
LSV3	M92/2010 *(year 2010)*	MG918125	LSV3/BombusBT35/2017 *(28 August 2017)*	MH350871	557	99.28%
LSV3	LSV3/CBNA/2017 *(10 August 2017)*	MH350882	LSV3/BombusBP9/2017 *(10 August 2017)*	MH350883	557	98.74%
LSV2	LSV2/286/2017 *(4 September 2017)*	MH350890	LSV2/BombusBP21/2017 *(22 August 2017)*	MH350889	557	99.64%

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
