# Peer review of "Determination of Genetically Identical Strains of Four Honeybee Viruses in Bumblebee Positive Samples"

_viruses, 2020, doi:10.3390/v12111310_

Round 1
Reviewer 1 Report
Since wild pollinator decline is a global problem, and honeybee viruses are probably one of the main factors of this decline, this study is valuable, but, in my opinion must be improved.
- The Abstract is somewhat "messy" and lacks a certain thought flow, so I would suggest linking the information contained in it with each other, so the reader doesn't experience a certain thought jump.
- to date there are 32 bee viruses discovered
- I am missing two recent important publications:
a) Beaurepaire, A.; Piot, N.; Doublet, V.; Antunez, K.; Campbell, E.; Chantawannakul, P.; Chejanovsky, N.; Gajda, A.; Heerman, M.; Panziera, D.; Smagghe, G.; Yañez, O.; de Miranda, J.R.; Dalmon, A. Diversity and Global Distribution of Viruses of the Western Honey Bee, Apis mellifera. Insects 2020, 11, 239.
b) Yañez O, Piot N, Dalmon A, de Miranda JR, Chantawannakul P, Panziera D, Amiri E, Smagghe G, Schroeder D and Chejanovsky N (2020) Bee Viruses: Routes of Infection in Hymenoptera. Front. Microbiol. 11:943. doi: 10.3389/fmicb.2020.00943
4. I am a bit confused about the bees collected between 2007 and 2018. When you used them to check for viruses, did you pick the ones from the same locations that bumblebees were collected? But most importantly, in my opinion, those bees are unnecessary in this paper as you collected extra bees at the same time at those same locations, so for me this should be compared, not the "historical bees".
5. As a general note, I have noticed a lot of mental shortcuts through the entire text and I suggest fixing that
Author Response
Re: Manuscript viruses-982737
On behalf of all my co-authors, I would like to thank you for the favorable
comments and constructive suggestions about our manuscript viruses-982737.
All corrections were clearly shown in the manuscript (in red color) and the responses to reviewers’ comments appear below (in blue color).
Responses to the comments of Reviewer 1:
R1: Since wild pollinator decline is a global problem, and honeybee viruses are probably one of the main factors of this decline, this study is valuable, but, in my opinion, must be improved.
Thank you for your comments. Below you will find how all your suggestions were implemented.
- R1: The Abstract is somewhat "messy" and lacks a certain thought flow, so I would suggest linking the information contained in it with each other, so the reader doesn't experience a certain thought jump.
Your suggestion was accepted. Two sentences at the beginning of the abstract and the last part of the abstract were deleted from the manuscript. The abstract is now presenting only the results of our study.
- R1: to date there are 32 bee viruses discovered
Your suggestion was accepted and add to the introduction chapter in the manuscript.
- R1: I am missing two recent important publications:
- a) Beaurepaire, A.; Piot, N.; Doublet, V.; Antunez, K.; Campbell, E.; Chantawannakul, P.; Chejanovsky, N.; Gajda, A.; Heerman, M.; Panziera, D.; Smagghe, G.; Yañez, O.; de Miranda, J.R.; Dalmon, A. Diversity and Global Distribution of Viruses of the Western Honey Bee, Apis mellifera. Insects2020, 11, 239.
- b) Yañez O, Piot N, Dalmon A, de Miranda JR, Chantawannakul P, Panziera D, Amiri E, Smagghe G, Schroeder D and Chejanovsky N (2020) Bee Viruses: Routes of Infection in Hymenoptera. Microbiol.11:943. doi: 10.3389/fmicb.2020.00943
Your suggestion was accepted and both recent publications from 2020 were included in the manuscript and in the reference chapter.
Beaurepaire A, Piot N, Doublet, V, Antunez K, Campbell E, Chantawannakul P, Chejanovsky N, Gajda A, Heerman M, Panziera D, Smagghe G, Yañez O, de Miranda JR, Dalmon A. Diversity and Global Distribution of Viruses of the Western Honey Bee, Apis mellifera. Insects 2020, 11, 239.
Yañez O, Piot N, Dalmon A, de Miranda JR, Chantawannakul P, Panziera D, Amiri E, Smagghe G, Schroeder D, Chejanovsky N. Bee Viruses: Routes of Infection in Hymenoptera. Front Microbiol 2020 11:943. doi: 10.3389/fmicb.2020.00943.
- R1: I am a bit confused about the bees collected between 2007 and 2018. When you used them to check for viruses, did you pick the ones from the same locations that bumblebees were collected? But most importantly, in my opinion, those bees are unnecessary in this paper as you collected extra bees at the same time at those same locations, so for me this should be compared, not the "historical bees".
Yes, you are right. The positive samples of honeybees were collected for sequencing: first 180 honeybee samples were collected from the whole territory of Slovenia within a voluntary monitoring program running between 2007 and 2018 and second on four locations at the same time as bumblebees were collected in August 2017 and 2018. The first group of positive samples is very important for understanding which strains of honeybee viruses are circulating on our territory in the last ten years. As shown on phylogenetic trees (Fig. 3, 4, 5, 6) several strains of viruses collected within the voluntary monitoring programs were genetically closely related to bumblebee positive samples (between 99 and 100 % nucleotide identity). This observation confirmed that endemically present viral strains circulating on one the same territory longer period than only one season and are regularly detected in both species.
As a supplementary material one table for this manuscript was prepared, where for all individual sequences additional data are presented: the exact date of sampling and if the sample was collected from one of four locations (location 1, 2, 3, 4) or within monitoring program.
- R1: As a general note, I have noticed a lot of mental shortcuts through the entire text and I suggest fixing that
Thank you for your comments. The manuscript was revised and we did our best to improve the manuscript.
Reviewer 2 Report
GENERAL
This is a nice addition to an important and current topic: the sharing of pathogens between managed and wild pollinators. The article is well written but has a few consistent English grammatical errors (individuum, the inappropriate use of 'the': simple things...) that could be easily ironed out with help of an English language expert. Overal though a nice descriptive study with no major faults.
INTRODUCTION
LSV may be the same virus as the historically described Bee virus X and Bee virus Y. See the BeeBook chapter on viruses for details (DOI 10.3896/IBRA.1.52.4.22).
Although most of these viruses were first detected in honeybees, that does not mean that honeybees are the only, or even primary host. Perhaps it would be more accurate to call these 'bee viruses' which would fit better with both our current understanding of these viruses and the context of the manuscript. Calling them 'honeybee viruses' sounds too species-specific.
MATERIALS & METHODS
Design derived from combining two different projects, which shows in the difference in sampling strategy for honeybees (pools of 50 or 10) and bumblebees (individual). This is not ideal but the data are handled conservatively and differences are unlikely to majorly affect the conclusions of the study, especially for the phylogenetic analyses. A pooled sample of 10 bees is more likely to be positive than that of a single bee, since only a few bees out of 10 need to be positive for the whole sample to become positive. The BeeBook chapter on statistics (DOI: 10.3896/IBRA.1.52.4.13) discusses in detail the effects of pooling in samples on detection. So the positive rates for the honeybees may be overestimated compared to those for the bumblebees. Since the detection rates are mostly used in a descriptive sense, and not for either modelling or statistical analyses, it has little effect on the conclusions of the paper which are in any case mostly focused on the sequence data. However, the authors should perhaps include a statement in their results, so that the readers are aware of the possible skew in the detection rates between honeybees and bumblebees.
RESULTS
See the comments above about comparing prevalences between honeybee and bumblebee samples.
Table 1 and Figure 2 cover the same data. Perhaps this information could be combined in a single figure instead, with separate histograms for the different bumblebee species. Then the table could go as supplementary data.
I am not sure what the difference is between sections 3.2 and 3.3. Both seem to emphasize that the nucleotide sequences of these viruses are identical or similar between bee species. These sections are also very repetitive, almost a sample-by-sample description of the nucleotide identities of the viruses detected. It does not really provide anything insightful that is anot also already provided by teh figures and tables. This could/should be shortened considerably.
Neither Table 2 nor Table 3 are necessary. Table 2 in particular contributes very little. Table 3 is simply a re-worked version of Table 1. They can go as supplementary information.
Figures 3, 4, 5 and 6 are the main highlight of the manuscript. The authors could help themselves and their readers by making these easier to understand, for instance by the use of colours and markers: to distinguish the different bee species, their geographic origins and the year of collection. There ar eplenty of examples in the literature that also deal with the phylogeny of biogeographic and temporal samples, where the use of colour, shades and blocking make a big difference to presentation and understanding. I would also recommend trying different tree shapes and topologies. The circular topology works well for ABPV, with very many near-identical isolates around the edge, but less well for BQCV, where perhaps a traditional square topology might be more illustrative.
DISCUSSION
This is also a bit repetitive, with the results section. The authors could have widened the scope by discussing the implications of their results (similar/different genetic strains in different geographies/bee species) for managed and wild bee health. There are now many studies that have done something similar, from which the authors can draw inspiration.
These are essentially qualitative results, both the prevalence and the sequence data. However, just because a virus is detected does not mean that it is necessarily deleterious for the bee. The consequences of these viruses for the different bee species is also highly dependent on how much of these viruses is found in the bees. There is a growing body of literature of the effects of these viruses on different bee species (DOI: 10.1016/j.jip.2014.06.011; doi: 10.1098/rsos.200480; Furst et al. for example) which might be useful to include in the discussion, as well as the multiple ecological roles of pathogens in general (doi: 10.1128/JVI.02974-14; doi.org/10.1016/j.cois.2019.03.004). Something worth thinking about.
Good luck!
Author Response
Re: Manuscript viruses-982737
On behalf of all my co-authors, I would like to thank you for the favorable
comments and constructive suggestions about our manuscript viruses-982737.
All corrections were clearly shown in the manuscript (in red color) and the responses to reviewers’ comments appear below (in blue color).
Responses to the comments of Reviewer 2:
R2: This is a nice addition to an important and current topic: the sharing of pathogens between managed and wild pollinators. The article is well written but has a few consistent English grammatical errors (individuum, the inappropriate use of 'the': simple things...) that could be easily ironed out with help of an English language expert. Overall though a nice descriptive study with no major faults.
Thank you for your favorable comments. Below you will find how all your suggestions were implemented.
INTRODUCTION
R2: LSV may be the same virus as the historically described Bee virus X and Bee virus Y. See the BeeBook chapter on viruses for details (DOI 10.3896/IBRA.1.52.4.22).
Your suggestion was accepted. The following text was added to the introduction and suggested BeeBook among references.
LSV has also been also detected with similar incidences and titres in historical European honeybee samples. LSV-1 and LSV-2 have strong similarities in capsid and genome size, seasonal incidence, predominantly adult-based infection, and absence of overt symptoms with Bee virus Y and Bee virus X respectively, and may therefore be related (De Miranda et al., 2012).
References
De Miranda JR, Bailey L, Ball BV, Blanchard P, Budge GE, Chejanovsky N, Chen Y-P, Gauthier L, Genersch E, de Graaf DC, Ribière M, Ryabov E, De Smet L & van der Steen JJM. Standard methods for virus research in Apis mellifera, J Apicult Res 2013; 52:4, 1-56. DOI:10.3896/IBRA.1.52.4.22
R2: Although most of these viruses were first detected in honeybees, that does not mean that honeybees are the only, or even primary host. Perhaps it would be more accurate to call these 'bee viruses' which would fit better with both our current understanding of these viruses and the context of the manuscript. Calling them 'honeybee viruses' sounds too species-specific.
Yes, you are right. Thank you for your suggestion. We add a few sentences into the discussion chapter as follows.
Although most of these viruses were historically first detected in honeybees, that does not mean that honeybees are the only or even primary host. Perhaps it would be more accurate in the future to call these ''bee viruses'' which would fit better with both our current understanding of virus infections in different bee species, because calling them ''honeybee viruses'' sounds too species-specific.
MATERIALS & METHODS
R2: Design derived from combining two different projects, which shows in the difference in sampling strategy for honeybees (pools of 50 or 10) and bumblebees (individual). This is not ideal but the data are handled conservatively and differences are unlikely to majorly affect the conclusions of the study, especially for the phylogenetic analyses. A pooled sample of 10 bees is more likely to be positive than that of a single bee, since only a few bees out of 10 need to be positive for the whole sample to become positive. The BeeBook chapter on statistics (DOI: 10.3896/IBRA.1.52.4.13) discusses in detail the effects of pooling in samples on detection. So the positive rates for the honeybees may be overestimated compared to those for the bumblebees. Since the detection rates are mostly used in a descriptive sense, and not for either modelling or statistical analyses, it has little effect on the conclusions of the paper which are in any case mostly focused on the sequence data. However, the authors should perhaps include a statement in their results, so that the readers are aware of the possible skew in the detection rates between honeybees and bumblebees.
Thank you for your suggestion. In chapter M&M we changed the number of collected samples of bumblebees from 50 to 10 honeybees because 10 samples were analyzed as a pool, not confusing reader with this numbers.
In the first, from 2007 to 2018, 180 samples of honeybee workers (10 honeybees in each sample) were collected within a voluntary monitoring program.
In the discussion chapter, we add following sentences.
We are aware that pooled samples of 10 bees are more likely to be positive than that of a single bumblebee since only a few bees out of 10 need to be positive for the whole sample to be positive. The positive rates for tested honeybee samples may be in this study overestimated compared to those results for testing single bumblebees. But the positive samples were mostly used in this study in a descriptive way and for phylogenetic studies and have little effect on the conclusions of this study.
RESULTS
R2: See the comments above about comparing prevalences between honeybee and bumblebee samples.
In the discussion chapter, we add following sentences (see above) to explain this for readers.
R2: Table 1 and Figure 2 cover the same data. Perhaps this information could be combined in a single figure instead, with separate histograms for the different bumblebee species. Then the table could go as supplementary data.
Thank you for your remarks. Your suggestion was accepted.
The data presented in table 1 is related to the prevalence of individual species in 2017 and 2018. From data, we can see the variations in prevalence between two collected seasons for individual species. To our opinion, this table 1 is more important for reader and we were decided to remain Table 1 in the manuscript.
In Figure 2, the summary results for the prevalence of six viruses detected on four locations in honeybees and bumblebee samples during 2017 and 2018 are presented. Although this figure presents important summary data in graphs for six viruses and variances for two consecutive years, this figure was removed into ’’Supplementary data 1’’ of the manuscript as was suggested.
R2: I am not sure what the difference is between sections 3.2 and 3.3. Both seem to emphasize that the nucleotide sequences of these viruses are identical or similar between bee species. These sections are also very repetitive, almost a sample-by-sample description of the nucleotide identities of the viruses detected. It does not really provide anything insightful that is anot also already provided by teh figures and tables. This could/should be shortened considerably.
Thank you for your observation. Your suggestion was accepted, now results are presented in one chapter
Chapter 3.2 describes the comparison of closely related viral strains detected between 2007 and 2018 within a voluntary monitoring program on the whole territory of Slovenia to the bumblebee samples (long time comparison) and also honeybee and bumblebee samples from four locations were compared.
R2: Neither Table 2 nor Table 3 are necessary. Table 2 in particular contributes very little. Table 3 is simply a re-worked version of Table 1. They can go as supplementary information.
Thank you for your suggestion.
Table 2 present representatives of most closely related strains with high nucleotide identity observed for ABPV, BQCV, SBV, and LSV among bumblebees (Bombus spp.) and honeybees (Apis mellifera) positive samples, confirming direct evidence of the same strains (ABPV, BQCV, SBV, and LSV), detected in both species of bees, with presented data of nucleotide identity-which is the core message of this manuscript. Probably this will be later confirmed also for several other viruses, which were not included in our study. So, we insist, that Table 2 is important for the presentation of the results of this and will remain manuscript.
Table 3 presents the percentage of detected positive samples for ABPV, BQCV, CBPV, DWV, SBV, and LSV by specific RT-PCR method for Bombus spp. collected in August 2017 and 2018 at four different geographic areas (Locations 1–4) in Slovenia. Here results are presented not only for four different locations and observed high variations between locations, but also between the year 2017 and 2018. Although these results are presented as ‘’average’’ in Table 1, can be important data for the reader, deeply understand, that high variation can exist, and that summary results sometimes cannot present important data, for this reason, Table 3 was prepared. Nevertheless, Table 3 will be moved into the supplement of this manuscript and presented as Supplementary data 3.
R2: Figures 3, 4, 5 and 6 are the main highlight of the manuscript. The authors could help themselves and their readers by making these easier to understand, for instance by the use of colours and markers: to distinguish the different bee species, their geographic origins and the year of collection. There ar eplenty of examples in the literature that also deal with the phylogeny of biogeographic and temporal samples, where the use of colour, shades and blocking make a big difference to presentation and understanding. I would also recommend trying different tree shapes and topologies. The circular topology works well for ABPV, with very many near-identical isolates around the edge, but less well for BQCV, where perhaps a traditional square topology might be more illustrative.
Thank you for your suggestion. Data for 158 new sequences of viruses (ABPV, BQCV, SBV, and LSV) were included in this study. This is a rather high number of sequences presented in one manuscript. The reason that the circular shape of phylogenetic was used because these trees required less space, but still important data can be visible. Important data for the reader are presented with names of each sample – see M&M description and some data (collection time, species, location, GenBank accession numbers) are also included and presented in supplementary data 2.
Figures 3, 4, 5, and 6 were changed as you suggested.
- Two markers and four colors were used for distinguishing species and different viruses.
- All four phylogenetic trees were presented in traditional square topology.
DISCUSSION
R2: This is also a bit repetitive, with the results section. The authors could have widened the scope by discussing the implications of their results (similar/different genetic strains in different geographies/bee species) for managed and wild bee health. There are now many studies that have done something similar, from which the authors can draw inspiration.
Thank you for your remarks. Several changes and adoption were made in the discussion chapter. All changes are written in red color.
R2: These are essentially qualitative results, both the prevalence and the sequence data. However, just because a virus is detected does not mean that it is necessarily deleterious for the bee. The consequences of these viruses for the different bee species is also highly dependent on how much of these viruses is found in the bees. There is a growing body of literature of the effects of these viruses on different bee species (DOI: 10.1016/j.jip.2014.06.011; doi: 10.1098/rsos.200480; Furst et al. for example) which might be useful to include in the discussion, as well as the multiple ecological roles of pathogens in general (doi: 10.1128/JVI.02974-14; doi.org/10.1016/j.cois.2019.03.004). Something worth thinking about.
Several new sentences were included in the discussion as suggested.
A series of experimental infections have shown that three viral types isolated from honeybees (DWV genotype A, DWV genotype B and BQCV) readily replicate within hosts of the bumblebee B. terrestris. Impacts of these honeybee-derived viruses - either injected or fed - on the mortality of B. terrestris workers were, however, negligible and probably dependent on host condition [25]. The identification and sequencing of the individual strain of honeybee viruses in bumblebee cannot be directly linked to the possible pathology but help us to understand the spillover events between different species. With experimental infection studies, acute virulence of individual honeybee viruses on bumblebees can be proved and is correlated with dose for the individual virus. By using the buff-tailed bumblebee, a generalist forager in the Palearctic region, previously have been demonstrated that oral feeding of 0.5 x 107 and 1 x 107 viral particles per bee of either Israeli acute paralysis (IAPV) or Kashmir bee virus, respectively, resulted in an active infection and fitness loss, while a dose of 0,5 x 106 IAPV particles per bee was not infectious [26]. Experiments elsewhere showed that oral infection of B. terrestris workers with 109 genome copies of a different honeybee virus, DWV, reduced the mean survival of B. terrestris workers by 6 days [6]. Nevertheless, when detecting genetically different field strains of viruses and the observed virulence data for only some of these viruses is known, the direct conclusions should not be generally accepted.
Although most of these viruses were historically first detected in honeybees, that does not mean that honeybees are the only or even primary host. Perhaps it would be more accurate in the future to call these ''bee viruses'' which would fit better with both our current understanding of virus infections in different bee species, because calling them ''honeybee viruses'' sounds too species-specific.
Our new understanding of the role of viruses in mutualistic symbiotic relationships with their hosts is expanding as our knowledge of the virome, through sequencing technologies and bioinformatic strategies is rapidly increasing. With an increasing number of detected and sequenced honeybee viruses, some aspects of how these viruses might be fused with their hosts in symbiogenetic relationships should be also considered. Viewing detected virus infections in the context of ecology provides a framework for a deeper understanding of the intertwined relationships of all life, including viruses. It is inevitable that more examples of mutualistic viruses will be revealed as we continue this exciting phase in the virology of virus discovery.
References
Six new references were added.
Beaurepaire A, Piot N, Doublet, V, Antunez K, Campbell E, Chantawannakul P, Chejanovsky N, Gajda A, Heerman M, Panziera D, Smagghe G, Yañez O, de Miranda JR, Dalmon A. Diversity and Global Distribution of Viruses of the Western Honey Bee, Apis mellifera. Insects 2020, 11, 239.
Yañez O, Piot N, Dalmon A, de Miranda JR, Chantawannakul P, Panziera D, Amiri E, Smagghe G, Schroeder D, Chejanovsky N. Bee Viruses: Routes of Infection in Hymenoptera. Front Microbiol 2020 11:943.
De Miranda JR, Bailey L, Ball BV, Blanchard P, Budge GE, Chejanovsky N, Chen Y-P, Gauthier L, Genersch E, de Graaf DC, Ribière M, Ryabov E, De Smet L & van der Steen JJM. Standard methods for virus research in Apis mellifera. J Apicult Res 2013; 52:4, 1-56.
Jamnikar Ciglenečki U, Toplak I. Genetic diversity of acute bee paralysis virus in Slovenian honeybee samples. Acta Vet Hun 2013; 61: 244-256.
Tehel A, Streicher T, Tragust S, Paxton RJ. Experimental infection of bumblebees with honeybee-associated viruses: no direct fitness cost but potential future threats to novel wild bee host. R Open Sci 2020 200480.
Meeus I, de Miranda JR, de Graaf DC, Wäckers F, Smagghe G. Effect of oral infection with Kashmir bee virus and Israeli acute bee paralysis virus on bumblebee (Bombus terrestris) reproductive success. J Invent Pathol 2014; 121: 64-69.
Thank you for your comments. The manuscript was revised, and we did our best to improve the manuscript.